# The Role of Neuroglia in Administrating Nerve Blockers and Anesthesia to Patients

**Anjali Patel** [1], **Raja Al-Bahou** [1], **Rajvi Thakkar** [1], **Drashti Patel** [1], **Devon Foster** [2], **Jonathan Benjamin** [1], **Marian Pedreira** [3] and **Brandon Lucke-Wold** [4,*]

1   College of Medicine, University of Florida, Gainesville, FL 32601, USA; anjalipatel@ufl.edu (A.P.); ralbahou@ufl.edu (R.A.-B.); r.thakkar@ufl.edu (R.T.); drashti.patel@ufl.edu (D.P.); jonathanbenjamin@ufl.edu (J.B.)
2   Herbert Wertheim College of Medicine, Florida International University, Miami, FL 32601, USA; devonfoster111698@gmail.com
3   Edward Via College of Osteopathic Medicine, Blacksburg, VA 24112, USA; mpedreira@vt.vom.edu
4   Department of Neurosurgery, University of Florida, Gainesville, FL 32601, USA
*   Correspondence: brandon.lucke-wold@neurosurgery.ufl.edu

**Abstract:** Dysfunction of the neuroglia can have profound consequences on the blood–brain barrier (BBB). Studies have shown that the disruption of astrocytic–endothelial interaction can compromise the permeability of BBB and its effectiveness in selectively regulating the exchange of substances. Microglia have recently been recognized to have a significant role in the initiation of chronic pain and in its interactions with various nerve blockers and anesthetic agents. Microglia have a role in pain resolution via a pathway that involves Cannabinoid receptor type 2 activation and MAP kinase phosphorylation. Understanding the role of these cells in the context of neuropathic pain and neurological disorders can aid in improving clinical outcomes and the challenging nature of managing pain. Advancing studies have proposed pharmacological and genetic modulation of microglia as a potential treatment option for patients with chronic pain.

**Keywords:** neuroglia; blood–brain barrier; MAP kinase phosphorylation; chronic pain

## 1. Introduction

The blood–brain barrier (BBB) plays a crucial role in protecting the central nervous system (CNS). The BBB is comprised of tightly sealed endothelial cells lining the blood vessels of the brain, which serve to strictly block the passage of undesired molecules into the brain. The role of the BBB is to safeguard what molecules, drugs, and nutrients are able to cross from the peripheral bloodstream into the brain tissue [1]. The semipermeable nature of the BBB selectively allows the passage of nutrients and desired molecules into the brain tissue and prevents the passage of potentially harmful toxins, pathogens, or autoantibodies [2].

Neuroglia constitute a diverse group of cells within the CNS, comprising astrocytes, microglia, oligodendrocytes, and ependymal cells [3]. While neurons are traditionally considered the primary functional units of the nervous system, neuroglia provide crucial support, maintaining the structural and functional integrity of the neural environment [3]. A comprehensive representation of all the various types of neuroglial cells is presented in Figure 1.

Astrocytes are the most abundant type of neuroglia and are particularly noteworthy for their intricate association with the BBB. They closely interact with the endothelial cells of the CNS and regulate the permeability of BBB by releasing various signaling molecules, including growth factors and cytokines, and maintaining the tight junctions in the blood vessels [4]. Astrocytes have a protrusion called "end-feet", which allows them to sheath brain vasculature to maintain tight junctions and regulate blood flow [5]. Moreover,

astrocytes play a role in regulating the transport of nutrients and waste products across the BBB, contributing to the overall homeostasis of the neural environment [6].

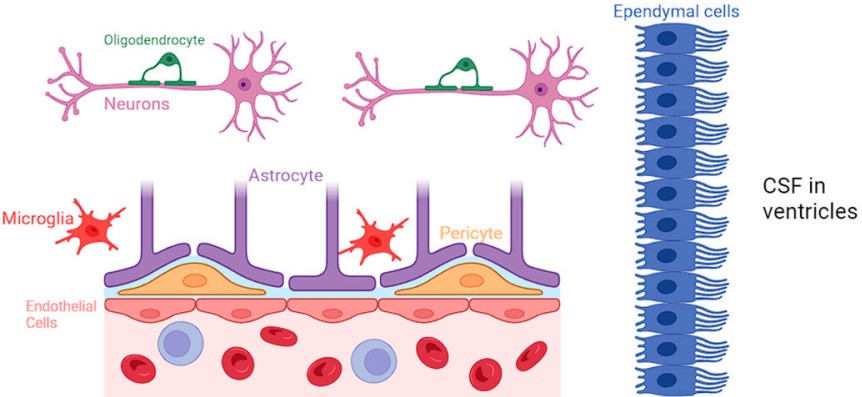

**Figure 1.** Biochemical specifics of the BBB and the specific role of each neuroglial cell.

Microglia are traditionally known for their immune surveillance and response functions. However, recent research has shed light on the intricate involvement of microglia in BBB regulation [7]. They actively participate in the modulation of tight junctions between endothelial cells, contributing to the selective permeability of the BBB. Studies have identified that following dysfunction of the BBB, microglia release factors such as transforming growth factor-beta (TGF-β) and interleukin-10 (IL-10), which have been shown to enhance the integrity of the BBB by promoting tight junction formation [8,9]. Microglia are also able to phagocytize cellular debris and pathogens that may be causing BBB disruption, and clearance of such materials can help return the BBB to a more normal state of functioning. Additionally, microglia can interact with astrocytes to promote the secretion of growth factors that can help heal damaged components of the BBB [10–12].

Oligodendrocytes play a pivotal role in the CNS by producing myelin and facilitating rapid electrical signal transmission. While the primary function of oligodendrocytes lies in myelination, emerging research suggests their involvement in BBB regulation. Studies have demonstrated that oligodendrocytes are closely associated with endothelial cells of the BBB, and they may participate in the modulation of barrier permeability. The interactions between oligodendrocytes and endothelial cells at the BBB highlight the complexity of neuroglial contributions to the maintenance of a tightly regulated neural microenvironment [13,14].

Ependymal cells contribute to the functionality of BBB primarily through their role in cerebrospinal fluid (CSF) production and circulation but also via their involvement in modulating the BBB's permeability. Recent studies have highlighted the expression of tight junction proteins in ependymal cells, indicating their potential contribution to the barrier function [15]. Ependymal cells can also work along with other glial cells, influencing the overall regulation of brain homeostasis.

Furthermore, pericytes serve the BBB by promoting blood vessel formation, maintaining the BBB, regulating immune cell entry, and controlling blood flow [16,17]. Pericytes stimulate angiogenesis through the secretion of vascular endothelial growth factor (VEGF), platelet-derived growth factor (PDGF), and angiopoietin-1, which can support the development of new capillaries. Additionally, pericytes control vasodilation and vasoconstriction, which allows them to control blood flow corresponding to the needs of the tissue. This helps tissue regenerate from damage and ensures the strength of the vasculature making up the BBB [18–20].

Dysfunction of the neuroglia can have profound consequences on the BBB. Studies have shown that the disruption of astrocytic–endothelial interaction can compromise the permeability of BBB and its effectivity on selectively regulating the exchange of substances between blood and the brain [7]. This compromised barrier function allows for the un-

controlled entry of potentially harmful molecules, including inflammatory mediators and toxins, into the brain parenchyma. This disruption can contribute to the development of neuroinflammatory diseases like multiple sclerosis (MS), neuromyelitis optica (NMO), and systemic lupus erythematosus (SLE) [2]. Dysfunction of the BBB plays a particular role in autoimmune neurological disorders due to effector molecules of the peripheral immune system being able to enter the brain and stimulate an inflammatory response, leading to disruption of normal neural function [21].

## 2. Pathophysiology of Pain

Pain is a nuanced and intricate experience, functioning as a major alarm system that signifies a potential threat or injury, which serves an adaptive role in protecting the body from harm. Yet, in certain cases, chronic pain can evolve into a maladaptive condition, causing significant personal and economic burdens [22–26]. Thus, effective pain management becomes crucial in mitigating these challenges, as it not only improves the individual's quality of life but also reduces the broader societal impact associated with healthcare costs, productivity loss, and emotional distress. However, the dynamics of pain involve a more complex interaction among biological, psychological, and emotional systems [27], and the perception of pain varies significantly among individuals, highlighting the high degree of inter- and intra-patient variability [28]. Recent advances in imaging modalities, such as functional magnetic resonance imaging (fMRI) and positron emission tomography (PET), have significantly enhanced our understanding of the central role played by the brain in the intricate processes of perceiving and modulating pain signals through the ascending and descending pathways.

These advances have also shed light on the interactions that macrophages have on our ability to feel pain. Many research studies have been conducted to analyze macrophages and their interactions with various nociceptors in our body. The Toll-like receptors that begin to circulate in our body after activation of nociceptors have been shown to activate macrophages and lengthen the pain that we feel. This is imperative to understanding when thinking about the differences between the ascending and descending pathways.

### 2.1. Ascending Pathway

Specialized neurons within the peripheral nervous system detect potentially harmful stimuli such as pain, temperature, and noxious chemicals through the activation of receptors like nociceptors [27,29]. These nociceptors transmit signals to the spinal cord, initiating the first leg of the ascending pathway [29]. Within the spinal cord, these signals synapse with secondary neurons that carry the information up to the brain, where the stimulus is further processed [29]. This process uses the ventrolateral system, alternatively termed the spinothalamic tract, which serves as a specialized tract in the transmission of pain and temperature signals to the thalamus—a central relay station for sensory information—and subsequently to the cortex of the brain [29,30]. As the signals ascend further, they reach various regions of the brain, including the somatosensory cortex, where the conscious perception of pain occurs [29,30]. Simultaneously, other brain areas, such as the limbic system, contribute to the emotional and affective aspects of pain [29]. The ascending pathways serve to relay information about the location, intensity, and quality of the pain, enabling the brain to generate an appropriate response [30].

### 2.2. Descending Pathway

In addition to the ascending pathways, the brain possesses a sophisticated system of descending pathways that play a crucial role in modulating the perception of pain. These pathways originate in higher brain regions, such as the periaqueductal gray (PAG) and the rostral ventromedial medulla (RVM) [22,30]. The PAG, in particular, acts as a key hub for pain modulation. Descending pathways exert both inhibitory and facilitatory influences on the transmission of pain signals [30,31]. Endogenous opioids, such as endorphins, are released in response to stress or injury, acting as natural pain relievers by binding to

receptors in the spinal cord and dampening the transmission of nociceptive signals [32]. Alternatively, descending pathways can also enhance pain transmission, emphasizing the dynamic and nuanced nature of pain modulation [30].

## 2.3. Advances in Pain Management

Chronic pain is maladaptive and remains a substantial burden for patients. In many cases, lifestyle modifications, as well as opiates, have been used as initial treatments. However, these approaches may prove ineffective or be linked to adverse systemic side effects, emphasizing the need for more precise and targeted interventional strategies. Thus, advancements in our understanding of pain perception and modulation allowed us to utilize different approaches, including ganglia injections, the role of calcitonin gene-related peptide (CGRP), and the involvement of Nav 1.8/1.7 channels, as promising targets in pain management, as shown in Figure 2.

**Figure 2.** Schematic of the various advancements in pain management.

## 2.4. Ganglia Injections

Ganglia injections are considered a promising approach when it comes to pain management. Sympathetic ganglia injections as a pain management approach have been described previously [33]. The mechanisms believed to contribute to sympathetic pain involve the impairment of inhibitory pain control as well as increased adrenergic excitability [33,34]. Thus, through the precise application of substances like local anesthetics (i.e., bupivacaine) or neuroleptic agents (i.e., alcohol) to specific ganglia, this method seeks to disrupt the positive feedback loop, thereby reducing central hyperexcitability and delivering a focused and targeted intervention [33]. Sympathetic blocks are increasingly used in managing both painful and nonpainful conditions, including postherpetic neuralgia [35], posttraumatic stress disorder (PTSD) [36], and hyperhidrosis [33]. Image-guided stellate ganglion injections have demonstrated effectiveness in addressing certain types of sympathetically maintained and visceral pain [35–40]. However, the lumbar sympathetic block is employed to address various types of pain, particularly those associated with the lower extremities, such as complex regional pain syndrome or sympathetic-mediated leg pain [41–43]. On the other hand, visceral abdominal and pelvic pain can be relieved by sympathetic blocks of the celiac plexus block [44–49], superior hypogastric [50–54], and ganglion impar [55–58]. This

method holds potential for various pain conditions, offering a more precise and minimally invasive alternative to traditional systemic treatments.

*2.5. Calcitonin Gene-Related Peptide*

CGRP, a neuropeptide primarily found in sensory nerves C and Aδ [59,60], has gained attention for its pivotal role in pain modulation [61]. Along with its vasodilatory action [62], recent research suggests that CGRP plays a crucial part in mediating pain signals through the facilitation of nociceptive transmission as well as peripheral and central sensitization [63–65]. Thus, therapies targeting CGRP receptors are being explored as potential treatments for chronic pain conditions. For instance, CGRP has been described in migraine pathology [66], and the subsequent development of CGRP antagonists was proven successful in treating acute episodes of migraine [67]. Furthermore, the later development of monoclonal antibodies against CGRP showed promise in treating chronic migraines [68–70]. Currently, three monoclonal antibodies (mAbs) have received approval for the prevention and management of migraines. Among these, two are designed to target the CGRP peptide, while the third is specifically directed toward the CGRP receptor [71]. Thus, blocking CGRP signaling may offer a novel avenue for pain relief without the side effects associated with traditional analgesics.

*2.6. NaV 1.8/1.7 Channels*

Also, Nav 1.8 and Nav 1.7 channels, specific sodium channels found in sensory neurons, have been identified as key players in pain transmission [72]. It is believed that the increased activation and/or expression of NaV1.7/1.8 channels have a role in the development and persistence of many forms of neuropathic pain [73]. Thus, since these channels contribute to the generation and propagation of pain signals, their modulation represents a promising therapeutic strategy. Even though agents selectively blocking Nav1.3, Nav1.7, Nav1.8, Cav3.2, and HCN2, as well as activators of Kv7.2, have demonstrated efficacy in alleviating indicators of neuropathic pain within animal models, their translation to clinical application has yet to prove successful [73–75]. Certain compounds fall short of achieving therapeutic endpoints, while others exhibit dose-limiting side effects, thus impeding their successful clinical implementation. Efforts to develop selective inhibitors for NaV 1.7 and NaV 1.8 channels are underway, aiming to provide effective pain relief while minimizing adverse effects on other physiological processes [74].

The experience of pain is a multifaceted phenomenon intricately woven into the fabric of human perception. The interplay between ascending and descending pathways within the brain illustrates the sophisticated mechanisms that contribute to the sensation, interpretation, and modulation of pain. A comprehensive understanding of these pathways is crucial not only for unraveling the mysteries of pain but also for developing targeted interventions to alleviate suffering and improve the quality of life for individuals experiencing pain. Thus, the exploration of ganglia injections, CGRP, and Nav 1.8/1.7 channels as mechanisms in pain modulation signifies a growing understanding of the intricate processes involved in pain perception. These emerging avenues hold promise for the development of targeted and effective interventions for patients with acute or chronic pain.

## 3. Introduction to Various Nerve Blockers Used in Anesthesia

The peripheral nerve blocks terminate pain signals the cerebral cortex receives from the spinal cord. Perioperative anesthetic nerve blocks can manage pain after procedures and reduce the need for postsurgical opioid consumption [76] when administered along with general anesthesia or autonomously in less complex surgeries using ultrasound-guided techniques [77]. Diagnostic nerve blocks in chronic pain can determine the anatomical source of pain signals and provide therapeutic utility [78]. Nerve blocks can reduce inflammation and provide temporary pain relief for acute and chronic upper and lower extremity pain. Damage to a sympathetic nerve chain can be used as a target for sympathetic nerve blocks when autonomic function damage and sympathetically mediated pain (SMP)

occurs. Stellate ganglion blocks further identify upper limb, head, and neck region nerve damage and block neural connections, improving the blood supply of the region and reducing adrenal hormone plasma concentration [79]. To diagnose facet joints as a source of pain, placebo-controlled zygapophysial blocks can be a cost-effective alternative to lumbar medial branch neurotomy [80]. We have summarized several types of common nerve blocks depending on the injury, clinical indications, and side effects in Table 1.

**Table 1.** Summary of the various types of common nerve blocks depending on the injury, clinical indications, and side effects.

| Procedure | Mechanism | Indication | Side Effects | Reference |
|---|---|---|---|---|
| Celiac plexus block (CPB) | Targets visceral afferent pain fibers from the liver, gallbladder, omentum, pancreas, mesentery, and stomach to the mid-transverse colon | Pain secondary to pancreatic cancer, chronic pancreatitis, and intractable abdominal pain | Transient or persistent diarrhea, paraplegia (anterior spinal artery syndrome), postural hypotension, pneumothorax | [81–83] |
| Epidural nerve block | Injected anesthetic in the epidural space temporarily numbs spinal nerves, blocking pain signals from spinal cord levels | Surgical procedures: pelvic fractures, cesarean delivery, labor analgesia, hepatic, gastric, and colonic surgeries Nonsurgical: myasthenia gravis, malignant hyperthermia, hyperreflexia | Hypotension, nausea, vomiting, post-puncture headache after dural perforation. [9] Incidence of transient paralysis is 0.1%; that of permanent paralysis is 0.02% [10]. Paresthesia with or without motor weakness, epidural hematoma, abscess, hypoalgesia of lower extremities | [84,85] |
| Genicular nerve block (GNB) | Anesthetizes sensory nerve terminal branches of genicular arteries or at the junction of the epiphysis and diaphysis of the femur and tibia, sparing motor function | Chronic knee osteoarthritis, post-operative knee pain, total knee arthroplasty, alternative to femoral, fascia iliaca, and adductor canal nerve blocks in knee injuries [11] | Leg muscle weakness, dizziness, and discomfort at injection site | [86–88] |
| Intercostal nerve block (ICNB) | Anesthetic injection to intercoastal nerves below each rib | Rib fracture neuralgia, thoracostomy analgesia, herpes zoster neuralgia, upper abdominal surgery, palliative cancer pain for rib and chest wall tumors | Self-limited bruising and soreness at the injection site. Serious: bleeding, infection, pneumothorax, nerve damage | [89,90] |
| Lumbar sympathetic nerve block | Disrupts the nerve supply from the preganglionic neurons exiting the spinal cord via the white rami of the ventral root of spinal nerves L1 to L4 and synapse at the lumbar sympathetic ganglion to the postganglionic neurons innervating the lower extremities | Sciatica, Complex Regional Pain Syndrome (CPRS), phantom limb pain, and lower limb painful ischemia | Flushing of skin, bleeding, bruising, soreness at the injection site, headache, and leg weakness on ipsilateral injection. Serious: infection, visceral injury, Horner's syndrome | [91–93] |

**Table 1.** *Cont.*

| Procedure | Mechanism | Indication | Side Effects | Reference |
|---|---|---|---|---|
| Occipital nerve block | C2 sensory neurons of the greater occipital nerve create a nociceptive pathway with the trigeminal nucleus caudalis, relieving compression and nerve irritation when targeted with an anesthetic | Occipital neuralgia, chronic intractable migraine, and cervicogenic and cluster headache treatment alternative in elderly and pregnant populations | Dizziness, vertigo, numbness, lightheadedness, vasovagal syncope, facial edema, and alopecia at injection if administered with steroid | [94,95] |
| Pudendal nerve block | Transcutaneous (perineal) or transvaginal approach targets the pudendal nerve trunk and its sensorimotor innervation | Pudendal neuralgia, obstetric (e.g., second stage of vaginal birth, vaginal repairs, hemorrhoidectomy), and urologic procedures (e.g.,transrectal ultrasound-guided prostate biopsy, transurethral prostatectomy) | Discomfort at the injection site, serious side effect of bladder and rectum structural injury, and pudendal artery puncture infection | [96,97] |
| Stellate ganglion block | Interrupts signals to the cervical sympathetic chain and postganglionic fibers for sympathetic innervation of upper limbs | CRPS of head and upper limbs, peripheral vascular disease, chronic post-surgical pain, postherpetic neuralgia, orofacial pain, scleroderma | Temporary pain, eyelid droopiness, fever, local blood aspiration, hematoma formation, spondylitis, and rare convulsions | [98,99] |
| Trigeminal nerve block | The ophthalmic (V1), maxillary (V2), and mandibular (V3) divisions and their corresponding nerves are blocked | Trigeminal neuralgia, pre-emptive analgesia in maxillofacial surgery | Difficulty chewing and swallowing and transient facial weakness and numbness | [100,101] |

*Impact of Lack of Neuroglia in Neurologic Disorders*

The dysfunction or absence of neuroglial cells is associated with the development of certain neurological diseases. Astrocytes are neuroglial cells associated with several central nervous system diseases when damaged or absent. Chronic migraines are a disease characterized by astrocyte dysfunction [102]. Patients with chronic migraines have headaches for at least 15 days in a month, with eight of those migraines fulfilling migraine criteria [103]. The relationship between astrocyte dysfunction and the development of chronic migraines may have to do with calcitonin gene-related peptide (CGRP). CGRP acts on astrocyte receptors to modulate neuropathic pain. The interaction of CGRP on astrocyte receptors causes Histone H3 lysine 9 acetylation, which is associated with inflammatory gene expression [104]. Pain can be experienced as a result of the response to the inflammatory processes by glial cells.

In addition to migraines, Amyotrophic Lateral Sclerosis (ALS) is another disease associated with neuroglial dysfunction. ALS is a neurological disease characterized by upper and lower motor neuron damage. With time, this disease progressively leads to muscle atrophy and paralysis. Microglial cells are involved in the pathogenesis of ALS. The early stages of ALS are associated with a decrease in the number of microglial cells. Interestingly, one study by Gerber et al. showed that the number of astrocytes is not altered at early symptomatic stages, but their intraspinal repartition is modified at symptom onset. [105,106]. In addition to the association of microglia with ALS pathogenesis, microglia also have a role in pain modulation. Neuromodulators produced by microglia can affect synaptic pruning, inducing pain after tissue or nerve injury. In contrast, microglia have a

role in pain resolution via a pathway that involves Cannabinoid receptor type 2 activation and MAP kinase phosphorylation [107]. Although pain is not a characteristic finding in ALS, the decreased number of microglial cells may affect their pain-modulation abilities.

Multiple sclerosis (MS) is another neurological disorder associated with glial cell dysfunction. MS is characteristically defined as a CNS demyelinating disorder due to an autoimmune response against myelin. This autoimmune response is typically associated with T-cells and B-cells, but microglia have been suspected to play a role as well. Activation of microglia leads to the release of cytotoxic nitrous oxide and superoxide radicals, causing CNS injury [108]. Additionally, oligodendrocyte injury is noted to occur in MS. A recent study supports a relationship between oligodendrocyte absence and pain after finding that genetic oligodendrocyte ablation rapidly triggers sensory changes that resemble central neuropathic pain [109,110]. This may explain the chronic pain symptoms experienced by some individuals with MS.

Similar to MS, Charcot–Marie–Tooth (CMT) disease is a demyelinating disorder of the nervous system. However, CMT differs in how it occurs due to dysfunction of the Schwann cells, which are located in the peripheral nervous system. Patients with CMT frequently experience pain or musculoskeletal deformities. More than one-third of patients with CMT manage their pain with analgesics such as NSAIDs or Acetaminophen [111]. Some patients may elect to undergo surgery to treat deformities and pain in their feet. The amount of studies reporting peripheral nerve blocks in patients with CMT is scarce, but one study evaluated the analgesic effect of a catheter-based sciatic nerve block in patients with CMT for postoperative pain control. The authors of this study concluded that peripheral nerve block in patients with CMT is safe and effective [112]. This is a reassuring finding, as patients with pre-existing neuropathies may experience complications from anesthesia [113].

Several of the discussed neurological diseases associated with glial cells have microglial dysfunction in common. As mentioned above, microglia have recently been recognized to have a significant role in the initiation of chronic pain. Advancing studies have proposed pharmacological and genetic modulation of microglia as a potential treatment option for patients with chronic pain. Even though there are no FDA-approved drugs that specifically target microglia, there are some available medications that have a degree of microglia modulation. These drugs can be used as analgesics for certain chronic pain syndromes [114]. Patients with neurological disorders involving microglial dysfunction may have unexpected effects with microglial-modulating drugs. Therefore, these drugs may not be an option for these individuals.

## 4. Future Developments in Alternative Nerve-Blockage Therapies

Botulinum neurotoxins (BoNTs) have been used for many years in patients with neuropathic pain [115]. BoNTs exert their effect by blocking the release of certain neurotransmitters, such as glutamate, substance P, and CGRP. Some of these neurotransmitters are associated with the activation of glial cells. Therefore, BoNTs also play a role in pain modulation by inhibiting the activation of glial cells associated with chronic pain [116].

Recent advances in the treatment of chronic neuropathic pain with paresthesia-free side effects include Spinal Cord Stimulation (SCS). Although this therapy suppresses central neuron excitability and causes a reduction in pain scores, it is associated with serious adverse effects of hematoma secondary to dural punctures. SCS is preferred over deep brain stimulation or motor cortex stimulation due to its moderate efficacy and minimal side effects [117]. Peripheral nerve stimulation has shown success in treating acute post-surgical pain by applying current to large-diameter myelinated afferent fibers and interfering with central pain signals. PNS can provide advantages in comparison to epidural local anesthetic injections for short-term pain relief without risk of infection, local anesthetic exhaustion, urinary retention, or motor weakness [112].

Novel, minimally invasive, painless therapies for alternative neuropathic pain include Transcranial Direct Current Stimulation (tDCS) and Remote Electrical Neuromodulation

(REN) [118]. tDCS reduces overall pain intensity in diabetic polyneuropathy patients with MRI-guided targeted tissue ablation and magnetic stimulation of the motor cortex. [119] REN therapy is effective for the acute treatment of migraines by inducing conditioned pain modulation from upper arm peripheral nerves [120]. Immersive Virtual Reality, although challenging for clinicians with limited technological experience and expensive, was found by a preliminary study examining 3D mirror feedback therapy to significantly reduce chronic upper extremity neuropathic pain for a brief period [121]. Further investigations regarding the role of glial cell response to neuromodulation in IVR can identify target glial cells for other regions of pain. To prevent the degeneration of injured nerves and replace lost neural cells, stem cell regeneration with transplantation has been studied with GABAergic neuron transplants in an injured spinal cord and reduces clinical symptoms of hyperalgesia and spontaneous pain. [122] Understanding the role of neuroglia in the context of neuropathic pain and neurological disorders can aid in improving clinical outcomes and the challenging nature of managing pain.

There have been advances in various clinical trials for the potential to manipulate microglia for patients suffering from chronic pain. One drug called Minocycline is a microglial inhibitor that has been indicated for low back pain, but they are currently in the recruiting status of the study [114]. Another drug called Tetrahydrocannabinol is a cannabinoid-receptor agonist that serves to modulate pain. This clinical trial is currently active [114]. A completed clinical trial involves the use of low-dose Naltrexone (a TLR4 antagonist) for fibromyalgia [114]. However, this drug is still pending FDA approval. There was another clinical study filled with patients suffering from post-herpetic neuralgia [114]. Unfortunately, the effects of the drug to decrease pain had minimal impact on the majority of the patients.

In conclusion, the neuroglia plays an imperative role in administrating nerve blockers and anesthetic agents to patients. Our review focused on microglia having a role in pain resolution via various pathways, such as the activation of cannabinoid receptor type 2 and MAP kinase phosphorylation. We described the intricate relationship between cells that create the blood–brain barrier and the impact that anesthetic agents and nerve blocks have on them. Future studies do need to be conducted in order to further augment the science behind proposed pharmacological and genetic modulation of microglia as potential treatment options that can be offered to patients suffering from chronic pain.

**Author Contributions:** Conceptualization, A.P. and B.L.-W.; methodology, A.P. and B.L.-W.; validation, A.P. and B.L.-W.; formal analysis, A.P.; investigation, B.L.-W.; writing—original draft preparation, A.P., R.A.-B., R.T., D.P., D.F., J.B., M.P. and B.L.-W.; writing—review and editing, A.P., R.A.-B., R.T., D.P., D.F., J.B., M.P. and B.L.-W.; visualization, A.P.; supervision, B.L.-W. All authors have read and agreed to the published version of the manuscript.

**Funding:** This research received no external funding.

**Institutional Review Board Statement:** Not applicable.

**Informed Consent Statement:** Not applicable.

**Data Availability Statement:** No new data were created.

**Conflicts of Interest:** The authors declare no conflicts of interest.

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
