# Peer review of "The Role of Neuroglia in Administrating Nerve Blockers and Anesthesia to Patients"

_2571-6980, doi:10.3390/neuroglia5010002_

Round 1
Reviewer 1 Report
Comments and Suggestions for Authors
This review is of great interest and describes in detail the role of microglia in pain control through cannabinoid type 2 receptor activation and MAP kinase phosphorylation, and the role of microglia in their interactions with various nerve blockers and anaesthetic agents. The real highlight of this review concerns studies that have proposed pharmacological and genetic modulation of microglia as a potential treatment option for patients with chronic pain.
1) Introduction
This paragraph is clear, the cited references are mostly recent publications and relevant.
3) Introduction to various Nerve Blockers used in Anesthesia
This paragraph is well described and well explored.
4.) Future developments in alternative nerve blockage therapies
This section reviews the most recent studies that have proposed pharmacological and genetic modulation of microglia as a potential treatment option for patients with chronic pain.
This review is relevant and of great interest to the scientific community, clear, comprehensive and of relevance in the field of neuroglia and chronic pain, however, some considerations by the authors and conclusions are missing at the end of the review, in order to improve the quality of work.
The cited references are mostly recent publications and relevant. The figures are appropriate, easy to interpret and understand.
The author presents a detailed manuscript, that is of high quality, in particular the topic discussed is of great interest, original and I recommend its publication on neuroglia after some changes:
- some considerations by the authors and conclusions at the end of the review, it is necessary to add a paragraph 5 with the conclusions.
Author Response
Thank you for your kind and constructive feedback. We have added an additional paragraph to our conclusion. Please let us know if there are any other changes that should be made. Thank you.

Reviewer 2 Report
Comments and Suggestions for Authors
Overall the information presented represents valuable information regarding the roles of Neuroglia in administrating Nerve Blockers and Anesthesia to patients. I have a few suggestions
It will be good to explain the role of microglia in pain resolution (with pathway) with the help of a figure.
How do infiltrating macrophages affect the pain? Does it have a role in pain management?
Please give current status/ clinical trials regarding the modulation of microglia as a potential treatment option for patients with chronic pain.
Author Response
Hello! Thank you for providing us with kind and constructive feedback. I have attached the revisions. I hope you have a great day.
